# Economic Impact on Health and Well-Being: Comparative Study of Israeli and Japanese University "Help" Profession Students

Richard Isralowitz [1,*], Mor Yehudai [1], Daichi Sugawara [2] , Akihiro Masuyama [3], Shai-li Romem Porat [1], Adi Dagan [1] and Alexander Reznik [1]

[1] Regional Alcohol and Drug Abuse Research Center, Ben Gurion University of the Negev, Beer Sheva 84105, Israel

[2] Faculty of Human Services, University of Tsukuba, Tsukuba 305-8577, Ibaraki, Japan

[3] Faculty of Psychology, Iryo Sosei University, Iwaki 970-8551, Fukushima, Japan

[*] Correspondence: richard@bgu.ac.il; Tel.: +972-525-793009; Fax: +972-8-6428141

**Abstract:** Background: Deteriorating economic conditions caused by rising inflation and living expenses can have negative consequences for university students. This comparative study examined Israeli and Japanese "help" profession (e.g., medicine, nursing, social work, and psychology) students' fear of such conditions and its impact on their health and well-being. Methods: Data were collected from a cross-sectional sample of 848 university students from Israel and Japan (78.9% female, 20.4% male, and 0.7% other) during a 3-month period of economic decline in 2022. Reliable data-collection instruments and SPSS (version 25) were used for the study. Results: Overall, Japanese students evidenced a higher level of economic well-being than their Israeli counterparts. This finding may have been a result of the lower inflation and living costs in Japan. However, most survey respondents evidenced a fear of deteriorating economic conditions that was significantly associated with psycho-emotional behavior, including increased burnout, substance use, unhealthy food intake, weight gain, and resilience regardless of gender and religiosity. Conclusions: The study findings showed the impact of deteriorating economic conditions on the health and well-being of "help" profession students. These results are preliminary; however, they do serve as an early warning of the key challenges that may need to be considered and addressed for prevention and intervention purposes. Further research should be conducted in other countries and over different time periods to substantiate present findings.

**Keywords:** university students; economic concern; well-being; stress; burnout; loneliness; substance use; eating behavior

The importance of higher education worldwide has sparked a growing interest in factors associated with students' well-being and success (Feldman et al. 2016). For many young adults, higher education is a time of transition and adjustment to meet social, academic, and financial demands (Briggs et al. 2012; Medlicott et al. 2021; Richardson et al. 2017). However, such changes may threaten student psychological well-being and mental health. A student who is unable to adjust and apply coping techniques against stressors may encounter negative emotions, troublesome situations, and loneliness (Diehl and Hilger 2016; Hicks and Heastie 2008; Kılınç et al. 2020). Studies of mental health difficulties among university students suggest that such issues may worsen throughout an academic degree program. While the worsening of symptoms may not be caused by higher education itself, it has been suggested that daily stressors associated with living conditions are a contributing factor (Cvetkovski et al. 2019; Duffy et al. 2020; Macaskill 2013; Medlicott et al. 2021).

The World Health Organization declared a public health emergency of international proportion in early 2020. Since that time, institutions of higher education, faculty members, and students have experienced considerable challenges, including those of an economic nature. COVID-19 and its variants remain a serious health concern and a significant impact

factor associated with deteriorating economic conditions, with inflation affecting living conditions, food, work, and study among students.

For many college students, a limited income and rising tuition and living costs are stressors that may affect student mental health and well-being (Barbayannis et al. 2022). Psychological factors associated with coping in a difficult living environment are well documented. For example, in 2009, unemployment was at 10%, with a scarcity of manufacturing, construction, and retail jobs; countries worldwide were burdened with debt that could not be repaid. Then as now, the impact on people of all ages, including university students (Greenglass et al. 2013), affected mental health and performance (Carey et al. 2021; Martinez 2020). Furthermore, the termination or reduction of health and social programs in response to economic conditions affect young people in academia preparing for a career in the "help" professions. Far-reaching, long-term consequences include psychological distress, depression, anxiety, lower social functioning and mental health, drug overdose, and death (Hammarström and Virtanen 2019; Li and Toll 2021; McIntyre and Lee 2020; Ofoegbu et al. 2020; Schwandt 2019). In a scoping review, Guerra and Eboreime (2021) reported increased depressive symptoms, self-harming behavior, and suicide. Additionally, because many students need to work to cover education and living costs, the loss of a job may be a factor associated with increased consumption of alcohol, tobacco, and other harmful substances (Bocchino et al. 2021).

Loneliness is common among many groups and is mostly short-lived. Among students, loneliness has been found to be associated with chronic stress as well as depression, affecting academic functioning and quality of life (Konstantinov et al. 2022; Stickley and Ueda 2022; Vanhalst et al. 2015); suicidality; and suicide mortality (Brafman et al. 2021; McIntyre and Lee 2020; Tzouvara et al. 2015). Furthermore, university student loneliness has been found to be associated with intensive media use, including on-line gambling and gaming (Aalbers et al. 2019); difficulty initiating and maintaining sleep (Hayley et al. 2017); and academic failure (Stadtfeld et al. 2019).

## 1. University Students in the "Help" Professions

Studies have shown a relationship between the adverse effects of economic crises and the mental health of "help" (i.e., medicine, nursing, allied health services, social work, and psychology) care professionals and students, of which the majority are female (Economou et al. 2013; Madianos et al. 2014; Reznik et al. 2022a, 2022b; Sher 2020; Yehudai et al. 2020). Deteriorating economic conditions can have deleterious effects on the psycho-emotional behavior, wellbeing, and success of those preparing for such careers (Berg-Cross and Green 2009; Frasquilho et al. 2015; Gestsdottir et al. 2021; Hammarström and Virtanen 2019; Li and Toll 2021; Negash et al. 2021). Furthermore, those on the front lines of service provision are likely to experience considerable burdens due to disaster conditions, resulting in increased vulnerability to anxiety, stress, and insomnia that can lead to emotional exhaustion, depression, and other mental health issues (Alshekaili et al. 2020; Gong et al. 2022; Repon et al. 2021; Shreffler et al. 2020).

This study examined the impact of the present uncertain economic conditions, in part precipitated by the pandemic, on the health and well-being of Israeli and Japanese "help" profession students. We hypothesized that students' concern about such conditions and its impact on their psycho-emotional behavior are associated with country, gender, and religiosity.

## 2. Methods

Established in 1996, the Regional Alcohol and Drug Abuse Research (RADAR) Center at the Ben Gurion University of the Negev has received recognition and award from the US National Institute on Drug Abuse for its contributions to scientific diplomacy through efforts in international collaborative research. For this study, the RADAR Center partnered with an Israeli non-profit organization whose primary aim is to collect data and promote the health and well-being of university students throughout the country. In Japan, university student data were collected at two universities—the University of Tsukuba and Iryo Sosei

University. Specifically, data were collected from "help" profession students in relation to their concern regarding economic conditions and its impact on their psycho-emotional well-being, including resilience, burnout, loneliness, substance use, and eating behavior. The Qualtrics software platform was used for the survey.

Five scales were used for data collection to address the following issues: (1) economic concern (ECS) was assessed via the COVID-19 fear scale (Ahorsu et al. 2020); (2) economic well-being (EWBS) was evaluated using 10 quality-of-life indicators (Tulsky et al. 2015); (3) resilience (BRS) was assessed in terms of the ability to bounce back or recover from stress (Smith et al. 2008); (4) burnout (SBM) was assessed according to the respondent's level of physical, emotional, and mental exhaustion (Malach-Pines 2005); and (5) loneliness (SLS) of an emotional and social nature was evaluated as a "feeling of missing an intimate relationship (emotional loneliness) or wider social network (social loneliness)" (Gierveld and Tilburg 2006). Furthermore, students were asked about their substance use (i.e., tobacco, alcohol, cannabis, and prescription drugs) and eating behavior before and after COVID-19.

The survey instruments used were translated into Hebrew and Japanese; then, they were back-translated into English by a multi-national team of researchers to ensure that the content and vocabulary were appropriate. The Cronbach reliability scores of the survey instruments used were: ECS = 0.885, EWBS = 0.889/0.916 (before and after COVID-19), BRS = 0.818, SBM = 0.921, and SLS = 0.688. Additionally, the Israeli and Japanese investigators received approval from the ethics committees of the universities involved to ensure that the appropriate steps were taken to protect the rights and welfare of the survey participants. These ethics approval processes are equivalent to established regulations to help protect the rights and welfare of human research subjects (Breault 2006). No external grant funding was received for the study. Respondents were advised that the survey was compliant with all ethical standards, their responses would remain confidential, and responses to the survey constituted consent to participate.

### 2.1. Statistical Analysis

All statistical analyses were conducted using SPSS, version 25. The Pearson's chi-squared test and the Mann–Whitney test were applied for qualitative/nonparametric variables, whereas the t test, correlation analysis, and one- and two-way ANOVA were used for continuous variables. Stepwise regression analysis was used to identify possible key economic concern predictors associated with Israeli and Japanese "help" profession students' psycho-emotional behavior.

### 2.2. Participants

A total of 848 "help" profession (i.e., medicine, nursing, psychology, and social work) students completed the online survey from January to July 2022. Respondent details were as follows: 60.4% (*n* = 512) Israeli and 39.6% (*n* = 336) Japanese; 20.4% male, 78.9% female, and 0.7% other; 67.7% secular; and 39.1% married/partnered. Table 1 provides information about the respondent demographic characteristics.

**Table 1.** Demographic characteristics.

| | Country | | | |
|---|---|---|---|---|
| | **Total** **(*n* = 848)** | **Japan** **(*n* = 336)** | **Israel** **(*n* = 512)** | $\chi^2$ **or *t*-Test** ***p*-Value** |
| Gender, % (*n*) | | | | |
| Male | 20.4 (173) | 28.9 (97) | 14.8 (76) | |
| Female | 78.9 (669) | 71.1 (239) | 84.0 (430) | <0.001 |
| Other [1] | 0.7 (6) | 0.0 (0) | 1.2 (6) | |

**Table 1.** *Cont.*

| | Country | | | |
|---|---|---|---|---|
| | **Total** (*n* = 848) | **Japan** (*n* = 336) | **Israel** (*n* = 512) | **χ² or *t*-Test** *p*-Value |
| Age: | | | | |
| Mean (SD) | 23.9 (5.5) | 19.8 (1.3) | 26.6 (5.5) | |
| Median | 23.0 | 20.0 | 25.0 | <0.001 |
| 95% CI | 23.6–24.3 | 19.7–20.0 | 26.2–27.1 | |
| Religiosity, % (*n*) | | | | |
| Secular | 67.7 (573) | 82.7 (278) | 57.7 (295) | <0.001 |
| Non-secular | 32.3 (274) | 17.3 (58) | 42.3 (216) | |
| Marital status, % (*n*) | | | | |
| Married/partnered | 39.1 (327) | 25.6 (86) | 48.2 (241) | <0.001 |
| Other | 60.9 (506) | 74.4 (250) | 51.8 (259) | |

[1] The category "other" was excluded from all analyses based on gender.

## 3. Results

The mean age of the respondents was 23.9 years (SD = 5.5), with the Israeli students being older than the Japanese students (t(842) = 6.831; *p* < 0.001). The mean economic concern (ECS) scores were higher among Japanese than Israeli students (26.3 (SD = 8.3) vs. 25.0 (SD = 5.5); t(792) = 2.562; *p* = 0.011). Regardless of country, economic concern was higher among females than males (26.1 (SD = 7.3) vs. 24.7 (SD = 6.9); t(788) = 2.095; *p* = 0.036). No significant differences were found among the respondents based on their religiosity and marital statuses. However, the two-way ANOVA results evidence a significant difference in ECS values based on gender and country: (F1786) = 4.724; *p* = 0.030) (Figure 1).

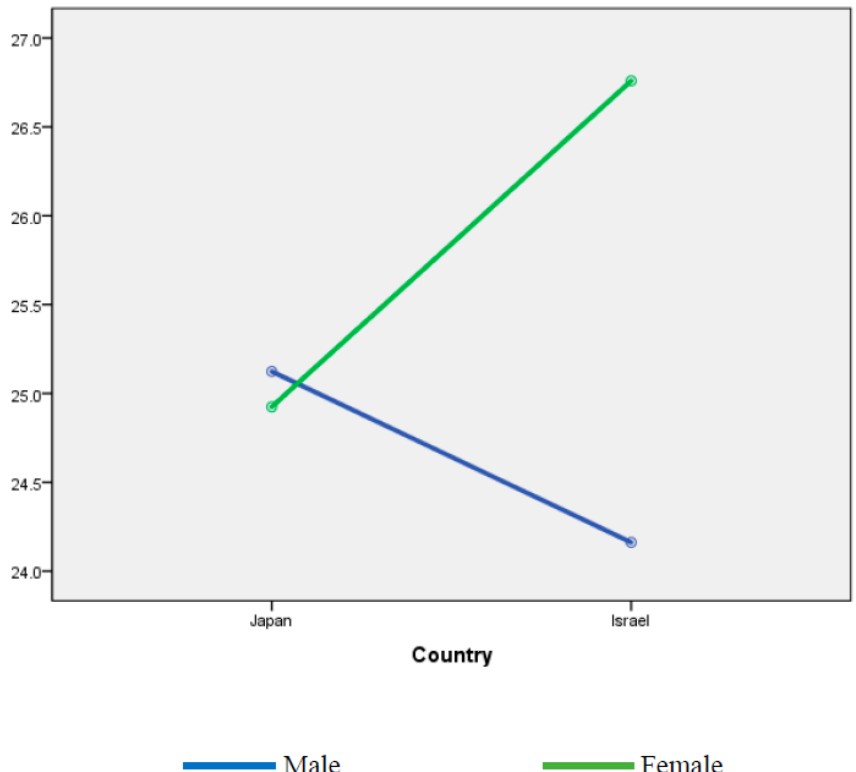

**Figure 1.** Economic concern by country and gender.

The overall economic well-being (EWBS) mean score before COVID-19 was 34.4 (SD = 7.1), being higher among Israeli compared to Japanese students (36.0 (SD = 6.9) vs.

32.3 (SD = 7.0); t(765) = 7.402; $p < 0.001$) and among married/partnered students compared to those who were not (35.5 (SD = 6.7) vs. 33.7 (SD = 7.4); t(764) = 3.351; $p = 0.001$). However, current (after COVID-19) EWBS scores were higher among Japanese compared to Israeli students (31.2 (SD = 7.5) vs. 28.7 (SD = 8.5); t(710) = 4.100; $p < 0.001$) and among those who reported that they were secular compared to those who were religious (30.3 (SD = 7.8) vs. 28.9 (SD = 8.7); t(709) = 2.161; $p = 0.031$). Overall, the results evidenced a significant decrease in economic well-being after the COVID-19 pandemic (t(704) = 13.711; $p < 0.001$). The two-way ANOVA results evidenced no significant difference in current well-being values based on country and gender (F(1703) = 1.165; $p = 0.281$), country and religiosity (F(1707) = 0.914; $p = 0.339$), and country and marital status (F(1707) = 0.162; $p = 0.687$).

Resilience (i.e., BRS scores) was found to be higher among Israeli compared to Japanese students (19.3 (SD = 3.9) vs. 17.2 (SD = 4.2); t(745) = 7.164; $p < 0.001$) and among males compared to females (19.0 (SD = 3.9) vs. 18.2 (4.2); t(740) = 2.021; $p = 0.044$). No significant differences were found in resilience scores based on religiosity and marital status. The two-way ANOVA results evidenced no significant difference in resilience values based on country and gender (F(1738) = 3.174; $p = 0.075$), country and religiosity (F(1742) = 1.464; $p = 0.227$), and country and marital status (F(1741) = 0.520; $p = 0.471$).

Burnout (i.e., SBM scores) was reported to be higher among Israeli compared to Japanese students (29.5 (SD = 8.9) vs. 26.4 (SD = 9.1); t(737) = 4.253; $p < 0.001$). Female (28.9 (SD = 9.1) vs. 25.6 (SD = 8.6); t(432) = 4.183; $p < 0.001$) and non-secular (29.6 (SD = 9.0) vs. 27.6 (SD = 9.1); t(736) = 2.850 $p = 0.004$) students reported more burnout. No significant differences were found based on marital status. The two-way ANOVA results evidenced no significant difference in burnout values based on country and gender (F(1730) = 0.812; $p = 0.366$), country and religiosity (F(1734) = 0.915; $p = 0.339$), and country and marital status (F(1733) = 0.094; $p = 0.760$). Correlations were found between the values of economic concern and those of resilience -0.118 ($p = 001$) and burnout 0.468 ($p < 0.001$).

Loneliness (i.e., SLS scores) tended to be more prevalent among Japanese compared to Israeli students (Mann–Whitney U = 52797.0; $p < 0.001$), males compared to females (U = 37,501.0; $p = 0.021$), and non-married students compared to those who were married/partnered (U = 50,346.0; $p = 0.001$). Loneliness values from 0 to 6 were divided into three categories: 0–1, no or low loneliness; 2–4, medium loneliness; and 5–6, high loneliness. Table 2 provides information about the relationship between respondents' loneliness and their economic concern, well-being, resilience, and burnout values.

**Table 2.** Loneliness level by economic conditions, resilience, and burnout.

| | Level of Loneliness | | | |
|---|---|---|---|---|
| | No/Low (*n* = 143) | Medium (*n* = 313) | High (*n* = 254) | One-Way ANOVA *p*-Value |
| Economic concern, mean (SD) 95% CI | 22.7 (6.5) 21.6–23.8 | 26.1 (7.5) 25.2–26.9 | 27.3 (6.2) 26.5–28.0 | <0.001 |
| Economic well-being before COVID-19, mean (SD) 95% CI | 36.1 (7.1) 34.9–37.3 | 34.7 (7.2) 34.0–35.6 | 32.6 (6.8) 31.8–33.5 | <0.001 |
| Current economic well-being, mean (SD) 95% CI | 32.8 (7.6) 31.6–34.1 | 29.8 (8.3) 28.9–30.8 | 28.5 (7.5) 27.6–29.4 | <0.001 |
| Resilience, mean (SD) 95% CI | 20.7 (3.8) 20.1–21.4 | 18.2 (4.0) 17.8–18.7 | 16.9 (4.0) 16.5–17.4 | <0.001 |
| Burnout, mean (SD) 95% CI | 22.8 (7.6) 21.6–24.1 | 27.4 (8.2) 26.5–28.3 | 31.7 (8.6) 30.7–32.8 | <0.001 |

Substance use in the previous month (i.e., tobacco, alcohol, cannabis, or prescription drugs) was reported by 51.1% of the respondents, being more common among Israeli compared to Japanese students (73.5% vs. 24.1%; $\chi^2(1) = 179.446$, $p < 0.001$). Specifically,

respondents reported 19.3% tobacco, 45.0% alcohol, 12.4% cannabis, 8.1% pain reliever, and 7.5% sedative use. A significant difference in binge drinking (i.e., five or more drinks on a single occasion) was reported based on country, with Japanese (10.1%) scoring higher than Israeli (3.8%) students ($\chi^2(1) = 12.405$, $p < 0.001$). No significant differences were found for substance use in the previous month based on gender and religiosity. Married/partnered students reported a higher level of substance use in the previous month than those who were single (59.4% vs. 46.0%; $\chi^2(1) = 12.627$, $p < 0.001$). Due to economic conditions, 15.3% of the respondents reported an increase in substance use in the previous month, with more Israeli than Japanese students reporting this behavior pattern (19.8% vs. 9.8%; $\chi^2(1) = 14.200$, $p < 0.001$). The use of each substance increased by the following percentages: tobacco—7.3%; alcohol—7.5%; cannabis—3.5%; pain relievers—2.2%; and sedatives—4.2%. The two-way ANOVA results evidenced a significant difference in ECS values based on country and substance use in the previous month (F(1730) = 7.916; $p = 0.005$) (Figure 2) and country and above-usual substance use in the previous month (F(1743) = 35.980; $p < 0.001$) (Figure 3).

Overall, 30.6% of the student respondents reported depression, 54.0% exhaustion, 29.4% loneliness, 40.4% nervousness, and 33.2% anger. Additionally, 63.9% of the students reported psycho-emotional deterioration because of their concern about economic issues— more so among Israeli (75.5% vs. 49.4%; $\chi^2(1) = 55.332$, $p < 0.001$), female (67.2% vs. 52.5%; $\chi^2(1) = 12.036$, $p = 0.001$), non-secular (73.1% vs. 59.7%; $\chi^2(1) = 12.667$, $p < 0.001$), and married/partnered (71.2% vs. 59.6%; $\chi^2(1) = 10.356$, $p = 0.001$) students. Table 3 provides information about the relationship between respondents' substance use and their economic concern, resilience, and burnout values.

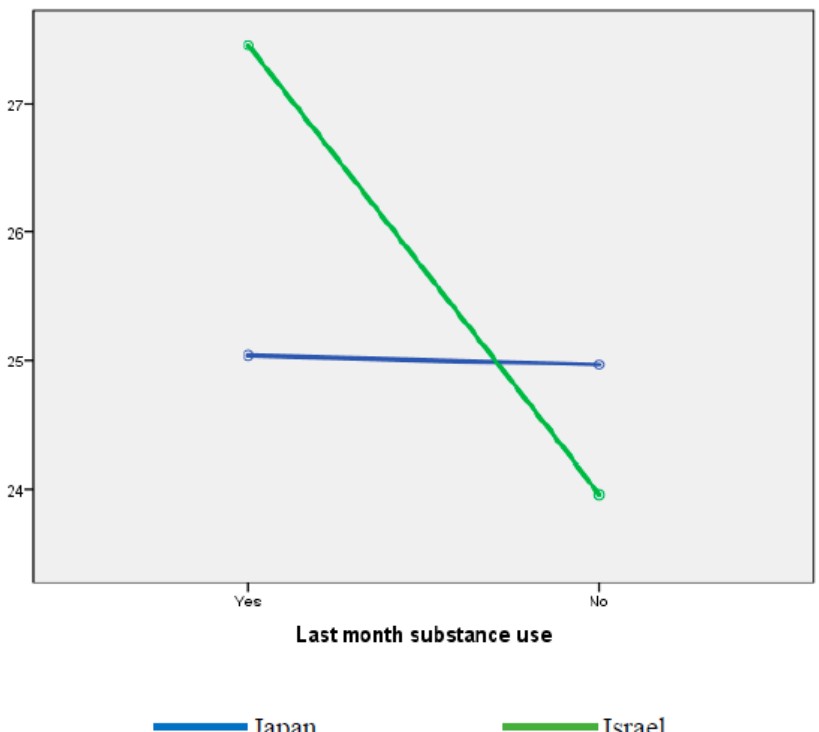

**Figure 2.** Economic concern by country and substance use in the previous month.

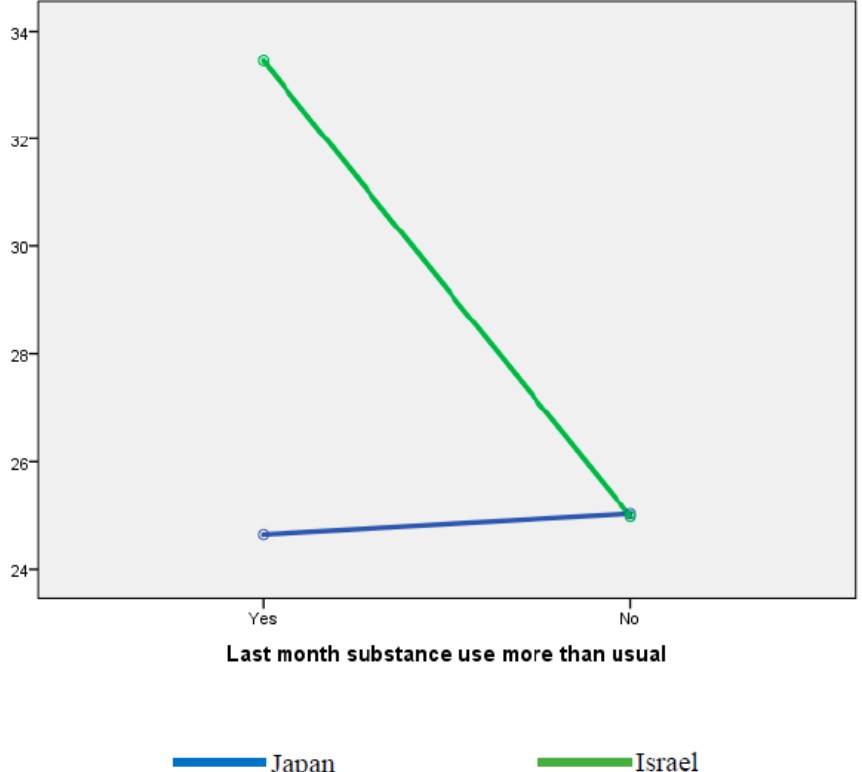

**Figure 3.** Economic concern by country and above-usual substance use in the previous month.

**Table 3.** Substance use and psycho-emotional well-being by economic concern, economic well-being, resilience, and burnout.

| | Any Substance Use in the Previous Month | | | Deterioration in Psycho-Emotional Well-Being | | |
|---|---|---|---|---|---|---|
| | Yes (*n* = 380) | No (*n* = 363) | *t*-Test *p*-Value | Yes (*n* = 484) | No (*n* = 273) | *t*-Test *p*-Value |
| Economic concern, mean (SD) 95% CI | 26.9 (8.0) 26.1–27.8 | 24.7 (6.0) 24.0–25.3 | <0.001 | 28.2 (6.8) 27.3–28.5 | 21.7 (6.0) 21.0–22.4 | <0.001 |
| Economic well-being before COVID-19, mean (SD) 95% CI | 35.6 (7.4) 34.8–36.3 | 33.0 (6.7) 32.3–33.7 | <0.001 | 34.3 (7.3) 33.6–34.9 | 34.6 (7.0) 33.8–35.4 | 0.578 |
| Current economic well-being, mean (SD) 95% CI | 28.8 (8.5) 27.9–29.7 | 30.9 (7.5) 30.1–31.7 | <0.001 | 27.9 (8.1) 27.2–28.7 | 33.2 (7.1) 32.3–34.0 | <0.001 |
| Resilience, mean (SD) 95% CI | 19.0 (4.1) 18.6–19.4 | 17.7 (4.2) 17.2–18.1 | <0.001 | 18.1 (4.2) 17.7–18.4 | 18.9 (4.0) 18.4–19.4 | 0.007 |
| Burnout, mean (SD) 95% CI | 29.2 (8.6) 28.3–30.1 | 26.9 (9.2) 26.0–27.9 | 0.001 | 31.0 (8.7) 30.2–31.8 | 23.1 (7.5) 22.2–24.0 | <0.001 |

Among all respondents, 69.3% reported eating more salt- and sugar-loaded food because of COVID-19 and economic conditions, more so among Israeli (75.3% vs. 63.1%; $\chi^2(1) = 11.875$, $p = 0.001$), female (73.3% vs. 54.9%; $\chi^2(1) = 18.747$, $p < 0.001$), non-secular (75.0% vs. 66.7%; $\chi^2(1) = 4.630$, $p = 0.031$), and married/partnered (74.5% vs. 66.0%; $\chi^2(1) = 5.364$, $p = 0.021$) students. Among all respondents, 45.1% reported a weight increase, more so among Israeli (51.2% vs. 37.8%; $\chi^2(1) = 13.381$, $p < 0.001$) and female (47.3% vs. 37.5%; $\chi^2(1) = 4.858$, $p = 0.028$) students. Students who reported eating more salt- and sugar-loaded food had a higher level of concern about current economic conditions ($t(671) = 3.717$;

$p < 0.001$) and evidenced more burnout (t(674) = 3.538; $p < 0.001$) and lower economic well-being (t(673) = 2.305; $p = 021$). Furthermore, those concerned about current economic conditions had more weight gain (t(728) = 4.148; $p < 0.001$) and burnout (t(734) = 5.838; $p < 0.001$) and lower economic well-being (t(708) = 3.675; $p < 0.001$). The two-way ANOVA results evidenced a significant difference in ECS values among the students based on country and unhealthy food intake (i.e., salt and sugar) (F(1669) = 10.452; $p = 0.001$), as well as a significant difference in current economic well-being based on country and unhealthy eating behavior (F(1651) = 5.109; $p = 0.024$).

Israeli and Japanese students' economic concern was examined as a dependent variable for stepwise regression analysis. Three significant predictors were found to explain the variable—burnout, psycho-emotional well-being deterioration, and an increase in substance use in the previous month. Burnout explained 21.1% of the variance (F(1665) = 178.6, $p < 0.001$), followed by psycho-emotional well-being deterioration (F(1664) = 56.8, $p < 0.001$) and an increase in substance use in the previous month (F(1663) = 20.6, $p < 0.001$). Other independent variables (i.e., country, gender, age, resilience, and marital status) did not significantly increase the proportion of explained variance. The resulting value of the explained variance (adjusted $R^2$) for the dependent variable (economic concern) was 0.293 (29.3%).

## 4. Discussion and Conclusions

This cross-sectional, comparative study examined the impact of economic concerns on Israeli and Japanese "help" profession students' well-being. Such students are or will be on the front-line of providing health and social services to people in need. The findings, significant and non-significant, tended to be consistent with those of other studies on the mental health and well-being of students under adverse conditions.

Our findings evidenced that economic concern affected Israeli and Japanese students' psychological well-being in terms of increased anxiety, depression, exhaustion, anger, loneliness, substance use, poor eating habits, and weight gain. These results tended to agree with those reported during the peak periods of COVID-19 infection (Isralowitz et al. 2022; Isralowitz and Reznik 2021; Karawekpanyawong et al. 2022; Konstantinov et al. 2022; Pavlenko et al. 2022; Sugawara et al. 2021; Zolotov et al. 2020).

The study findings showed that nearly 64% of those surveyed experienced a mental health deterioration because of current economic conditions. Female students, who are often the majority of those engaged in "help" professions, reported greater concern about the economic downturn; increased burnout, unhealthy food consumption, and weight gain; and less resilience compared to their male counterparts. However, they tended to be less lonely and less likely to report substance use in the previous month to cope with the current economic conditions.

The information generated from this survey and other studies about the negative impact of economic conditions on university students' well-being should be used for academic curriculum development, teaching, and prevention purposes. Furthermore, the information may contribute to promoting students' coping skills and management of their well-being. Relevant advice based on experience of coping under adverse conditions should perhaps be considered and made available to "help" profession students, online and in print, to mitigate stress and anxiety and prevent harmful behavior (Arcaya et al. 2020; Findley et al. 2016; Stephenson 2020; Zolotov et al. 2020).

The findings of this survey were limited to "help" profession students, who were mostly female, from two countries at one point in time under rapidly deteriorating economic conditions. Furthermore, this survey was limited because of the scope of factors examined. Further studies should include an examination of suicidality and suicide mortality (Gunnell et al. 2020; McIntyre and Lee 2020; Ryu et al. 2022; Sher 2020; Zalsman et al. 2020), as well as excessive online gaming (i.e., gaming disorders) and gambling during difficult times (Giardina et al. 2021; King et al. 2020; Wu et al. 2022).

**Author Contributions:** Conceptualization, R.I., M.Y., D.S., A.M., S.-l.R.P., A.D. and A.R.; methodology, R.I., A.R., D.S., A.M., M.Y. and A.D.; validation, A.R.; formal analysis, A.R.; research, R.I. and A.R.; resources, R.I., D.S. and A.M.; data curation, A.R., D.S. and A.M.; writing-original draft preparation, R.I., A.R. and S.-l.R.P.; writing—review and editing, R.I., A.R. and S.-l.R.P.; visualization, A.R.; supervision, R.I., A.R., D.S. and A.M.; project administration and editing, R.I., A.R., D.S. and A.R. All authors have read and agreed to the published version of the manuscript.

**Funding:** This research received no external funding.

**Institutional Review Board Statement:** The study was conducted in accordance with the Declaration of Helsinki, and approved by the Institutional Review Board (or Ethics Committee) of Ben Gurion University of the Negev, Israel; Faculty of Human Services, University of Tsukuba, Japan; and, Faculty of Psychology, Iryo Sosei University (Japan).

**Informed Consent Statement:** Informed consent was obtained from all subjects involved in the study.

**Acknowledgments:** We acknowledge the contribution made to this study by Toby and Mort (Z"L) Mower through their generous support of the Ben Gurion University of the Negev—Regional Alcohol and Drug Abuse Research (RADAR) Center.

**Conflicts of Interest:** The authors declare no conflict of interest.

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
