# Peer review of "Economic Impact on Health and Well-Being: Comparative Study of Israeli and Japanese University “Help” Profession Students"

_socsci, doi:10.3390/socsci11120561_

Round 1
Reviewer 1 Report
Reformat tables - left justify and add confidence intervals to figures.
I would like to see some discussion about what the economic circumstances are in Japan and Israel. This would add important contextual information.
Author Response
Response to the reviewer's comment
Reviewer's comment |
Response to the reviewer's comment |
Location within the manuscript applicable to the comment and revisions |
Reviewer’s #1 comment |
|
|
Reformat tables - left justify and add confidence intervals to figures. |
Done |
Table 1, p.6; Table 2, p.8; Table 3, p.11. |
I would like to see some discussion about what the economic circumstances are in Japan and Israel. This would add important contextual information. |
Brief mention is added in the abstract and in the text. Economic circumstances are unstable – interest rates for Israel during the last 12 months is 7.7%; in Japan it is 3.8%. It is noted in the paper that Japanese students reported a higher level of economic well-being than their Israeli counterparts. This finding may be a result of lower inflation and living costs in Japan. |
Abstract; p.7 |
Reviewer 2 Report
The authors analyze the economic consequences on the mental states and well-being of students of "help" professions.
The work is well-written and follows the standard structure of a scientific article. The subject matter is of heated relevance because many students are experiencing problems with depression and mental health. Not just in Israel and Japan, but around the world. It would be interesting to extend the analysis to other countries and compare them.
The empirical analyzes and results were presented correctly. The authors explain the work in detail. It might be interesting to extend the analysis with a quantile regression to determine the level of economic concern the covariates are explained.
The discussions and conclusions summarize and comment on the work done in a concise manner, which is much appreciated in these studies.
Minor comment:
- it would be better to introduce what the Cronbach reliability score indicates before showing the results. It would be more understanding.
Author Response
Reviewer’s #2 comment |
|
|
It would be interesting to extend the analysis to other countries and compare them. |
Agreed. This point has been mentioned in the abstract and Discussion/Conclusion. The authors welcome this, and will provide the data collection instrument for use elsewhere |
|
It might be interesting to extend the analysis with a quantile regression to determine the level of economic concern the covariates are explained. |
Additional information about the results of the regression analysis (what proportion of the explained variance is associated with each of the predictors) has been added to the text. Regarding quantile regression, all statistical analyses were conducted using SPSS, version 25. For quantile regression, higher SPSS versions are required. |
Results Section, p.12. |
it would be better to introduce what the Cronbach reliability score indicates before showing the results. It would be more understanding. |
Information about Cronbach alpha have been relocated to the Methods section. |
Methods Section, p. 5 |